# Impact of a Dietary Supplementation with French Maritime Pine Bark Extract Pycnogenol^®^ on Salivary and Serum Inflammatory Biomarkers During Non-Surgical Periodontal Therapy—A Randomized Placebo-Controlled Double-Blind Trial

**DOI:** 10.3390/nu17091546

**Published:** 2025-04-30

**Authors:** Jasmin Bayer, Nicole Karoline Petersen, Jeanine Veruschka Hess, Yvonne Jockel-Schneider, Petra Högger

**Affiliations:** 1Institute of Pharmacy and Food Chemistry, University of Würzburg, 97074 Würzburg, Germany; jasmin.bayer@uni-wuerzburg.de; 2Department of Periodontology, University Hospital Würzburg, 97070 Würzburg, Germany; petersen_n1@ukw.de (N.K.P.); hess_j1@uwk.de (J.V.H.); jockel_y@ukw.de (Y.J.-S.)

**Keywords:** Pycnogenol^®^, gingival inflammation, periodontitis, polyphenols, dietary supplementation

## Abstract

**Background**: Gingival inflammation is highly prevalent and may impact systemic health. While professional mechanical plaque removal (PMPR) is the standard treatment, dietary interventions may provide additional benefits. The French maritime pine bark extract Pycnogenol^®^ has anti-inflammatory and antioxidant properties, but its impact on inflammatory biomarkers in saliva and serum has not been studied in a controlled clinical trial. **Methods**: In this randomized, double-blind, placebo-controlled clinical trial, 91 participants received Pycnogenol^®^ (100 mg twice daily; *n* = 46) or a placebo (*n* = 45) following PMPR. Saliva and serum samples were collected at baseline, and after two and three months. Inflammatory biomarkers (IL-1β, IL-6, MMP-8, and MMP-9) and polyphenol concentrations were analyzed using ELISA and LC-MS/MS. **Results**: Pycnogenol^®^ supplementation significantly reduced salivary MMP-8 levels (*p* = 0.0261), and serum IL-6 levels compared to placebo (*p* = 0.0409). Additionally, ferulic acid, caffeic acid, and the gut microbial metabolite 5-(3,4-dihydroxyphenyl)-γ-valerolactone (M1) significantly increased in saliva following Pycnogenol^®^ intake. A correlation analysis revealed a significant inverse association between bleeding on probing and M1 concentration in saliva (r = −0.3476, *p* = 0.0167). **Conclusions**: Dietary supplementation with Pycnogenol^®^ significantly reduced key inflammatory biomarkers and increased polyphenol concentrations in saliva, suggesting a potential anti-inflammatory effect of Pycnogenol^®^ on gingival inflammation. **Trial registration**: ClinicalTrials.gov (NCT05786820).

## 1. Introduction

Periodontal health is characterized by the presence of a benign polymicrobial community encompassing, for example, *Proteobacteria*, *Firmicutes* phyla, and *Actinobacteria*. These bacteria form a symbiotic biofilm and support oral health by releasing antimicrobial compounds and inhibiting pathogenic colonization. The disruption of this host–microbe homeostasis entails the proliferation of pathobionts, for instance, *Porphyromonas gingivalis* or *Tannerella forsythia*, which trigger gum inflammation (gingivitis). The presence of the dysbiotic oral microbiome triggers a local host response, which, in turn, stimulates inflammatory cells, such as neutrophils, lymphocytes, and macrophages. These immune cells release various inflammatory mediators, including tumor necrosis factor-alpha (TNF-α), prostaglandin E2, interleukin (IL)-1, and IL-6. This leads to alveolar bone resorption due to the cytokine-mediated activation of osteoclasts and the release of matrix metalloproteinases (MMPs) [1,2,3]. Depending on the individual immune response, the inflammation of the soft tissue may be limited to the gums (gingivitis) or may progress and irreversibly affect the hard and soft tissue (periodontitis). Numerous studies have confirmed a shift in pro-inflammatory biomarkers in saliva as well as in the blood serum of diseased patients, underscoring their potential clinical utility for diagnosis, treatment, or treatment control [4,5,6].

Periodontal disease is globally highly prevalent. A recent cross-sectional epidemiological study in seven European countries concluded that the prevalence of oral conditions was higher than reported in previous literature. Gingival inflammation, determined by bleeding on probing (BoP), was observed at a minimum of one site in 87% of the participants. Gingivitis with BoP at more than 10% of the sites was recorded for 66% of the otherwise healthy adults [7].

Gingival inflammation is attenuated by local periodontal treatment in the course of professional mechanical plaque removal (PMPR). As an adjunct treatment, probiotic mouthwashes containing *Lactobacillus* species or *Bifidobacteria* have been explored [8]. Although probiotics do not eradicate pathobionts, their colonization in the oral cavity is modulated, supporting the host immune system and reducing the production of inflammatory mediators [2].

Gingival inflammation is also associated with increased oxidative stress elicited by reactive oxygen species released from, for example, activated neutrophils [9]. Patients with periodontitis were shown to have increased levels of free radicals and impaired antioxidant defense mechanisms. Supplementation with antioxidant compounds such as lycopene or vitamin E improved periodontal parameters [10]. In a double-blind randomized placebo-controlled study, supplementation with capsules containing concentrates of fruits, vegetables, and berry juice, BoP and the cumulative plaque score significantly improved in the group receiving fruit–vegetable capsules. Those capsules contained defined amounts of β-carotene, vitamin E, vitamin C, and folic acid. The content of polyphenols in the supplement was not analyzed due to the varying growth and harvest conditions of the fruits and vegetables [11]. However, it can be assumed that polyphenols contributed to the effects seen in the study participants. Polyphenols have antioxidant, anti-inflammatory, antimicrobial, and immunomodulatory effects [12], and beneficially interact with gut microbiota [13]. Polyphenol-containing plant extracts revealed the potential for the treatment or prevention of periodontal disease in vitro, ex vivo, and rodent in vivo studies [12]. More recently, multiple human trials with bioactive polyphenolic compounds have been reported. These studies applied the polyphenols topically as gels or mouthwashes [14].

To the best of our knowledge, no clinical trial perorally administering a polyphenol-containing plant extract to human patients with gingival inflammation has been reported. A dietary food supplement, a standardized extract from French maritime pine bark (Pycnogenol^®^) containing 65–75% oligomeric procyanidins and various small phenolic compounds such as taxifolin, ferulic acid, catechin, and protocatechuic acid, has been shown to inhibit alveolar bone resorption in a rat periodontitis model [15]. This extract has been broadly studied in human clinical studies [16], but not yet in gingival inflammation. Recently, we reported that constituents and a polyphenol metabolite produced by gut bacteria (5-(3,4-dihydroxyphenyl)-γ-valerolactone) were present in human saliva samples after the intake of Pycnogenol^®^ [17,18]. The purpose of the present study was to investigate the impact of a dietary supplementation with French maritime pine bark extract (Pycnogenol^®^) on gingival inflammation in a randomized, placebo-controlled, double-blind trial. The null hypothesis was that oral Pycnogenol^®^ supplementation has no effect on inflammatory biomarkers in saliva or serum compared to placebo during non-surgical periodontal therapy. In this manuscript, we focus on the concentrations of phenolic compounds in saliva and the levels of inflammatory markers in saliva and serum samples. Clinical results will be reported in a separate manuscript.

## 2. Materials and Methods

### 2.1. Study Design and Participants

This study was a randomized, double-blind, placebo-controlled clinical trial with a two-arm parallel-group design, registered in the ClinicalTrials.gov database (NCT05786820; 19 December 2022). It was conducted in accordance with the Declaration of Helsinki for medical research involving human subjects and received approval from the Ethics Committee of the University of Würzburg (260/19-me). A total of 91 subjects were enrolled within the Department of Periodontology at University Hospital of Julius-Maximilians-University and participated in the study between March 2022 and June 2024. All participants were provided with comprehensive information about the study and gave written consent prior to their involvement.

### 2.2. Intervention and Monitoring

The intervention in this study consisted of the administration of Pycnogenol^®^, a standardized extract of French maritime pine bark (Pinus pinaster Aiton) that meets the quality standards of the United States Pharmacopeia for dietary supplements, or a placebo. Participants were randomly assigned to one of the two groups in a 1:1 ratio. Randomization was performed using block randomization in Microsoft Excel (Version 2302, Microsoft Corporation, Redmond, OR, USA) with blocks of 10 participants. Random numbers were generated, ranked, and participants were assigned to either the Pycnogenol^®^ and placebo group accordingly. The study followed a double-blind design, ensuring that both participants and the involved dentists (Y.J-S., N.K.P., and J.V.H.) were blinded to group allocation. The blinding list was securely stored by the study evaluators (J.B. and P.H.). All participants underwent professional mechanical plaque removal (PMPR) at the beginning of the study, followed by daily intake of either Pycnogenol^®^ capsules (100 mg twice daily) or identically looking placebo capsules for three months.

During the study period, participants attended four scheduled visits. At the first visit (baseline), clinical, microbiological, and cardiovascular data were collected, participants were randomized, and the intervention started after the scheduled PMPR. At the second visit, two months after the start of the study, the clinical and microbiological parameters were collected again, as well as at the third visit. The third visit, which took place three months after the first assessment, marked the end of the intervention phase, and the fourth visit was three months after the end of supplementation to assess the long-term outcomes.

The sample size was calculated to achieve a statistical power of 80% and a significance level of 5%. Based on a standard deviation of SD = ±8.0% and a minimum detectable difference of ∆ = 5.0% of bleeding on probing (BoP), which was the primary objective of the study, the required sample size was determined to be 40 participants per group. To account for potential dropouts, a group size of 45 participants was pursued, resulting in a total of 90 participants.

Monitoring included adherence assessment by counting returned capsules and the documentation of intake in the participants diaries. The results of the salivary polyphenol levels and various inflammation-related biomarkers, analyzed at baseline, as well as two and three months after the start of the intervention, are reported here. Additional clinical parameters (e.g., bleeding on probing, and gingival index) will be detailed in a separate manuscript.

### 2.3. Selection Criteria: Inclusion/Exclusion Criteria

The selection criteria for participants in the study included specific inclusion and exclusion criteria. Adults between 35 and 85 years of age with a body mass index (BMI) between 20 and 30 kg/m^2^ and at least ten natural teeth were eligible to participate in the study. They had to have clinical signs of gingivitis, defined as at least 10% bleeding on probing (BoP) and a gingival index (GI) between 0 and 2 on at least three teeth. Exclusion criteria were the presence of inflammatory oral mucosal diseases other than gingivitis, a salivary flow rate below 0.1 mL/min, the inability to maintain regular oral hygiene, or physical or mental impairments that prevented compliance with the study protocol. Participants were excluded if they had a history of malignant disease, chemotherapy or radiotherapy in the past five years, were pregnant or breastfeeding, had acute infections such as HIV, or had metabolic bone disease. Patients were also excluded if they had taken antibiotics or anti-inflammatory drugs in the four weeks before screening, if they were taking drugs known to affect gingival inflammation (such as corticosteroids), or if they were undergoing active orthodontic treatment.

### 2.4. Biomarkers and Salivary Polyphenol Levels

Saliva and blood samples were collected to analyze specific biomarkers (IL-1β, IL-6, MMP-8, and MMP-9) and to quantify the concentration of seven polyphenols (taxifolin, ferulic acid, caffeic acid, gallic acid, trans para-coumaric acid, protocatechuic acid, and M1 (5-(3,4-dihydroxyphenyl)-γ-valerolactone)) in saliva. Saliva was collected using the SalivaBio Oral Swab (Biozol Diagnostica GmbH, Eching, Germany) by placing the swab under the tongue for approximately 5 min. Following collection, the swab was immediately stored at 4 °C before being centrifuged at 4000× *g* for 10 min at 4 °C. The collected saliva samples were aliquoted, rapidly frozen in liquid nitrogen, and stored at −80 °C until analysis. Blood samples were collected using the S-Monovette^®^ Serum Gel CAT (SARSTED AG & Co. KG, Nümbrecht, Germany) and stored at 4 °C immediately before centrifugation at 2500× *g* for 10 min to separate serum, which was aliquoted, frozen in liquid nitrogen, and stored at −80 °C until further analysis. To quantify the concentrations of the inflammatory biomarkers, enzyme-linked immunosorbent assay (ELISA) kits (IL-1 Beta EIA Kit, Salimetrics; Human MMP-9 ELISA Kit, Biorbyt; Human MMP-8 ELISA Kit, Biorbyt; Interleukin 6 ELISA Kit, Biorbyt; all procured from Biozol Diagnostica GmbH, Eching, Germany) were utilized, whereas a previously validated liquid chromatography tandem mass spectrometry (LC-MS/MS) method was employed to quantify the polyphenols [18].

### 2.5. Statistical Analysis

A range of statistical tests were employed to analyze the differences between the groups. All calculations were performed using GraphPad Prism Version 6.07 for Windows (Graphpad Software, La Jolla, CA, USA). Prior to the statistical tests, a two-tailed outlier test was conducted using the ROUT method in GraphPad Prism to identify and remove any outliers from the dataset. The differences in demographic characteristics were determined using *t*-test (for continuous variables) and Fisher’s exact test (for categorical variables). Polyphenol and biomarker concentrations within groups were analyzed using the Wilcoxon matched-pairs signed-rank test. The Mann–Whitney test and the repeated measures ANOVA were used for the comparison between the groups. To account for multiple post hoc comparisons, Sidak’s correction was applied. Furthermore, Fisher’s exact test was utilized to compare the number of participants for whom a particular polyphenol could be quantified. Correlation analyses were performed using Spearman’s rank correlation test. Significance was assumed at an α-level of <0.05.

## 3. Results

### 3.1. Participant Flow and Baseline Characteristics

The participant flow within the study in accordance with the Consolidated Standards of Reporting Trials (CONSORT) guidelines is illustrated in Figure 1. Initially, all potential subjects were screened for eligibility. After inclusion screening, 91 participants were randomized into two groups: 46 participants received Pycnogenol^®^, while 45 participants were allocated to the placebo group. A small number of participants withdrew from the study during its course (dropouts) in both groups. The data collected were then analyzed according to the group assignment.

The mean age was comparable in both groups (67.83 ± 9.18 years vs. 66.96 ± 8.20 years, *p* = 0.6348). However, there were significant differences in the gender distribution: the proportion of male participants in the Pycnogenol^®^ group was significantly higher than in the placebo group (58.7% vs. 26.7%, *p* = 0.0029). Other variables such as diet (vegetarian diet), smoking behaviour, and alcohol consumption did not differ significantly between the groups (Table 1).

### 3.2. Biomarkers and Salivary Polyphenol Levels

The analysis of the polyphenol concentrations in saliva and inflammation-related biomarkers in saliva and serum demonstrated significant differences between the Pycnogenol^®^ group and the placebo group. The mean values with standard deviation and *p*-values are presented in Table 2. It was evident from the outset that the baseline values of certain polyphenols differed between the two groups. Specifically, the baseline taxifolin concentrations were significantly higher in the Pycnogenol^®^ compared to the placebo group. Saliva concentrations of ferulic acid and caffeic acid substantially increased following Pycnogenol^®^ supplementation. These changes were particularly noticeable after two months, and the trend stabilized after three months. A similar observation was made for gallic acid concentrations, which increased significantly in the Pycnogenol^®^ group between baseline (45.59 ± 50.04 ng/mL) and two months (82.27 ± 84.34 ng/mL), while remaining relatively constant in the placebo group. The concentration of M1 was significantly higher in the Pycnogenol^®^ group compared to the placebo group after two months (1.67 ± 1.45 ng/mL vs. 0.08 ± 0.30 ng/mL), and this trend continued after three months (Figure 2).

In addition to the mean polyphenol concentrations, differences in the number of patients in whom the respective polyphenols were quantified were also analyzed. This showed that individual polyphenols were detected in more participants in the Pycnogenol^®^ group than in the placebo group. For instance, ferulic acid was detectable in an increasing number of participants from 11 to 17 in the Pycnogenol^®^ group, while a slight decrease in its presence was recorded over the study period in the placebo group (from 9 to 7 participants). In addition, the metabolite M1 was detected after two months in a significantly higher number of participants in the Pycnogenol^®^ group (13 participants) than in the placebo group (1 participant). However, the concentration of the other polyphenols, such as p-coumaric acid and protocatechuic acid, showed no significant differences between the groups.

In addition to the analysis of polyphenols, various biomarkers associated with inflammatory processes were also investigated. An overview of the trends in the individual biomarker concentrations between the two groups at the specified time points is shown in Figure 3. The concentration of salivary IL-1β non-significantly decreased from 214.69 ± 178.80 pg/mL to 157.22 ± 107.14 pg/mL in the Pycnogenol^®^ group over three months, while it remained largely unchanged in the placebo group (190.11 ± 161.39 pg/mL to 179.88 ± 132.07 pg/mL). A similar pattern was observed for MMP-8 in saliva, with significantly reduced levels (47.65 ± 33.39 ng/mL to 33.43 ± 23.15 ng/mL) in the Pycnogenol^®^ group after three months (*p* = 0.0261) compared to the placebo group (42.06 ± 28.58 ng/mL to 39.78 ± 29.69 ng/mL). In contrast, the concentrations for MMP-9 remained relatively constant across both groups, with a slight decrease (38.91 ± 15.37 ng/mL to 34.67 ± 11.69 ng/mL) observed in the Pycnogenol^®^ group (Table 2). At baseline, the serum IL-6 concentration was not significantly different between the two groups. However, after three months, significantly lower IL-6 levels were observed in the Pycnogenol^®^ compared to the placebo group (*p* = 0.0409). Within the Pycnogenol^®^ group, the IL-6 concentration revealed a minor decrease (44.20 ± 14.43 ng/mL to 43.16 ± 13.17 ng/mL), while an increase (48.88 ± 14.83 ng/mL to 51.27 ± 13.84 ng/mL) was recorded in the placebo group. Due to the significant gender imbalance between the Pycnogenol^®^ and placebo group, a subgroup analysis was conducted for the two biomarkers that showed statistically significant changes: salivary MMP-8 and serum IL-6. This analysis revealed no significant differences between male and female participants within either group at any time point. Given the absence of gender-related effects for these key outcomes, and to preserve statistical robustness, no additional subgroup analyses were performed.

### 3.3. Correlation Analysis

A Spearman rank correlation analysis revealed several significant associations between bleeding on probing (BoP) as a clinical parameter for gingival inflammation, inflammatory biomarkers, and salivary polyphenol levels (Table 3). BoP showed a weak but significant positive correlation with MMP-8 (r = 0.1906, *p* = 0.0203) and IL-1β (r = 0.2380, *p* = 0.0038), whereas a significant negative correlation was found between the BoP and M1 concentration in saliva (r = −0.3476, *p* = 0.0167; Figure 4). A further BoP evaluation will be presented in a separate manuscript.

Among the biomarkers, a strong positive correlation was observed between MMP-8 and IL-1β (r = 0.6979, *p* < 0.0001; Figure 4). Additionally, MMP-8 showed a significant correlation with IL-6 (r = 0.1922, *p* = 0.0201), while IL-6 demonstrated a weak but significant positive correlation with IL-1β (r = 0.1781, *p* = 0.0324).

An analysis of salivary polyphenols revealed several significant positive correlations. Ferulic acid showed a strong correlation with caffeic acid (r = 0.5108, *p* < 0.0001), while caffeic acid showed a significant positive correlation with both p-coumaric acid (r = 0.2946, *p* = 0.0031) and protocatechuic acid (r = 0.2608, *p* = 0.0141). In addition, protocatechuic acid showed a significant correlation with p-coumaric acid (r = 0.3053, *p* = 0.0012) and with gallic acid (r = 0.3780, *p* = 0.0048). Notably, gallic acid was also found to be significantly correlated with the gut metabolite M1 (r = 0.4945, *p* = 0.0087, Figure 4).

## 4. Discussion

In the present study, we investigated, for the first time, the impact of a procyanidine-rich dietary supplementation with French maritime pine bark (Pycnogenol^®^) on selected salivary and serum inflammatory biomarkers in a randomized placebo-controlled double-blind trial including 91 participants undergoing non-surgical periodontal therapy.

The French maritime pine bark extract Pycnogenol^®^ had been previously reported to beneficially influence the inflammatory processes in various disease contexts, such as osteoarthritis [16]. These effects have been attributed to the ability of its constituents and/or metabolites to inhibit the activation of the key inflammatory switch NF-κB and activity of cyclo-oxygenase (COX) enzymes, as well as to decrease the release of matrix metalloproteinases (MMPs), which play an essential role in tissue degradation and inflammation [16,19]. However, the impact of Pycnogenol^®^ on inflammatory biomarkers in oral inflammation had not been studied in a controlled human trial before.

In addition to professional mechanical plaque removal (PMPR), nutritional supplements have been investigated as adjunct approach in periodontal therapy. Several studies suggest that such dietary interventions may positively impact inflammatory processes [12,20,21,22]. Unlike traditional antimicrobials, which come with concerns regarding bacterial resistance, probiotics and dietary supplements may offer a safer alternative for reducing inflammation and restoring a healthy microbial balance [23,24]. Microbiological samples were collected as part of the study protocol and will be analyzed in further investigations to assess the possible shifts in the oral microbiota associated with Pycnogenol^®^ supplementation. Saliva has been shown to be a reliable source of microbial DNA and represents a non-invasive method suitable for microbial profiling [25]. Sugimoto et al. [15] demonstrated that Pycnogenol^®^ reduced alveolar bone resorption in a rat periodontitis model by inhibiting bacterial adhesion and osteoclastogenesis, suggesting potential benefits for restoring periodontal health. The present study is the first human study addressing the question of whether oral supplementation with Pycnogenol^®^ can positively influence inflammatory biomarkers after professional mechanical plaque removal. A previous study using chewing gum containing 5 mg Pycnogenol^®^ demonstrated significant reductions in gingival bleeding and plaque accumulation over two weeks, highlighting the potential of local application in managing gingival inflammation [26]. Further research may explore the use of Pycnogenol^®^ in surgical periodontal therapy, such as its incorporation into platelet-rich preparations or bioactive scaffolds [27]. However, oral administration was chosen in this study due to the pharmacokinetic profile of Pycnogenol^®^, including microbial metabolism and enterohepatic circulation, which support sustained systemic bioavailability and stable salivary concentrations [17].

Our results indicate that Pycnogenol^®^ supplementation led to significant reductions in key inflammatory biomarkers and increased concentrations of specific salivary polyphenols. These findings align with the previous research on the systemic anti-inflammatory effects of Pycnogenol^®^ [16].

MMP-8, a collagenase involved in periodontal tissue degradation, was significantly reduced in the Pycnogenol^®^ group compared to placebo. Given that elevated MMP-8 levels are associated with periodontal disease progression [28], its reduction suggests a potential protective effect of Pycnogenol^®^ on periodontal tissues. This finding is consistent with in vitro studies demonstrating that Pycnogenol^®^ inhibits matrix metalloproteinases [19]. A non-significant trend towards lower IL-1β levels in saliva was also observed, while IL-6 concentrations in serum were significantly reduced in the Pycnogenol^®^ group, further supporting its anti-inflammatory potential. Although the reductions in MMP-8 and IL-6 were statistically significant, the absolute changes were modest, and the clinical relevance of these biomarker shifts remains to be clarified.

A salivary polyphenol analysis revealed that Pycnogenol^®^ supplementation increased the levels of several polyphenols, including ferulic acid and caffeic acid. Notably, M1 (5-(3,4-dihydroxyphenyl)-γ-valerolactone), a gut-derived metabolite of Pycnogenol^®^ constituents, was significantly elevated in the Pycnogenol^®^ group compared to the placebo group. This increase not only confirms patient adherence to the supplementation protocol but also highlights M1 as a potential biomarker for compliance in future studies. Beyond its role in monitoring adherence, M1 may also function as a bioactive effector, contributing to the anti-inflammatory and antioxidant effects associated with Pycnogenol^®^ [16].

Correlation analyses further support these findings, revealing a significant inverse correlation between BoP and M1 (r = −0.3476, *p* = 0.0167), suggesting that M1 may play a role in reducing gingival inflammation. However, additional studies are needed to determine whether M1 contributes directly to the anti-inflammatory effects. In addition, BoP showed a significant positive correlation with MMP-8 (r = 0.1906, *p* = 0.0203) and IL-1β (r = 0.2380, *p* = 0.0038), indicating that higher levels of gingival bleeding were associated with increased inflammatory activity. Furthermore, the strong correlation between MMP-8 and IL-1β underscores the relevance of these biomarkers in periodontal disease progression. IL-1β is a key pro-inflammatory cytokine that promotes the activation of matrix metalloproteinases, including MMP-8, leading to extracellular matrix degradation and tissue destruction [29]. The observed association between these two markers reinforces the role of IL-1β in promoting periodontal inflammation by inducing MMP-8-mediated tissue breakdown, highlighting the potential benefits of Pycnogenol^®^ in modulating these inflammatory pathways.

Our findings are in line with the previous research on dietary interventions for periodontal health in animal models [15,30]. For instance, Laky et al. [30] demonstrated that quercetin, a polyphenol found in various foods, mitigated disease progression in experimental periodontitis.

This study represents the first controlled human trial assessing the effects of Pycnogenol^®^ on gingival inflammation, providing novel insights into its potential as an adjunctive therapy for periodontitis. The observed reductions in inflammation-related biomarkers and the correlation between BoP and M1 suggest that Pycnogenol^®^ may influence periodontal health through anti-inflammatory mechanisms.

However, the three-month administration of Pycnogenol^®^ may not fully reflect its long-term effects. Future research should investigate whether these effects persist over extended periods and whether Pycnogenol^®^ can contribute to sustained periodontal stability.

### Limitations of the Study

The present study has some limitations. We report no detailed clinical outcomes, such as bleeding on probing (BoP) or pocket probing depth (PPD). These parameters were systematically assessed during the trial but will be presented and discussed in a separate manuscript in order to maintain the specific biochemical focus of the current work. Including both datasets in a single publication would have exceeded the intended scope of this manuscript.

Furthermore, inflammatory biomarker levels were measured in saliva rather than in gingival crevicular fluid (GCF), which, more specifically, reflects the local periodontal environment. Saliva was chosen as the primary matrix because it allowed the simultaneous quantification of polyphenols and provided a sufficient sample volume for the biochemical analyses conducted. To minimize participant burden in this trial, we decided against an additional, more invasive sampling procedure such as GCF collection.

Another limitation is that participants with varying stages of periodontal disease were included, as the primary inclusion criterion was the presence of gingival inflammation (BoP ≥ 10%), regardless of a formal diagnosis of gingivitis or periodontitis. While this heterogeneity may have introduced some variability in the baseline inflammatory status, it reflects the real-life clinical spectrum of periodontal conditions typically encountered in practice. Moreover, all participants underwent standardized non-surgical periodontal therapy (PMPR) prior to the intervention, which helped to reduce the disease-related variability and allowed for meaningful comparisons of inflammatory biomarkers across the study population.

Additionally, although a significant gender imbalance was present between the groups, subgroup analyses for the two significantly altered biomarkers (MMP-8 and IL-6) revealed no relevant differences between male and female participants. Nevertheless, the possibility of gender-related biological variability cannot be entirely excluded and should be addressed in further studies with more balanced cohort compositions.

## 5. Conclusions

In conclusion, our study demonstrates that Pycnogenol^®^ supplementation significantly reduced key inflammatory biomarkers in saliva and serum while significantly increasing salivary polyphenol concentrations. The detection of the Pycnogenol^®^ gut microbial metabolite 5-(3,4-dihydroxyphenyl)-γ-valerolactone (M1) in saliva and its association with clinical improvements further support the relevance of these findings. These results provide a basis for further clinical research to explore Pycnogenol^®^ as an adjunctive therapy in periodontal treatment strategies.

## Figures and Tables

**Figure 1 nutrients-17-01546-f001:**
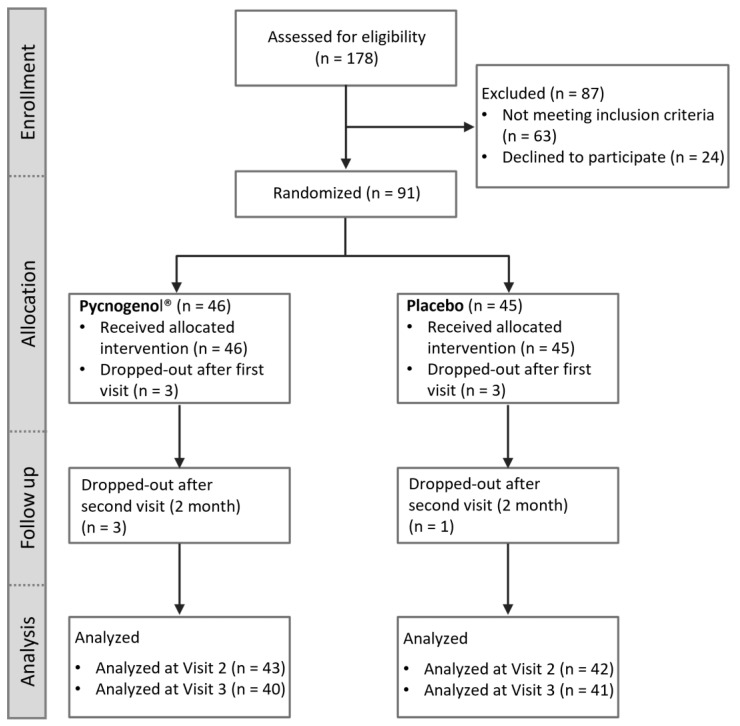
Consolidated Standards of Reporting Trials (CONSORT) flow chart illustrating the participant flow through the study, including enrollment, allocation to intervention groups (Pycnogenol^®^ and placebo), follow-up, and analysis.

**Figure 2 nutrients-17-01546-f002:**
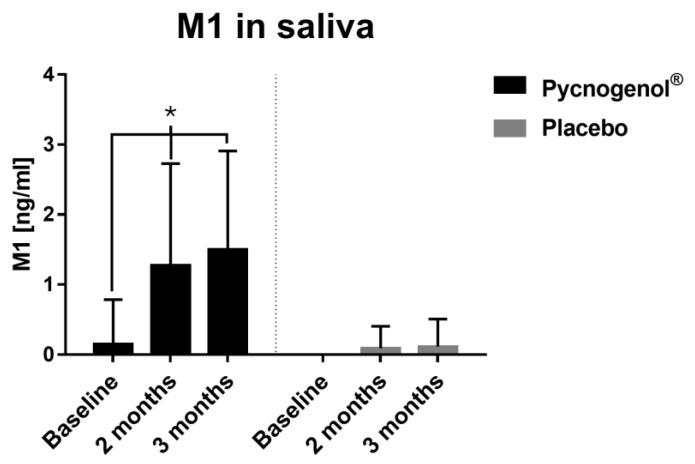
Saliva concentrations of M1 in the Pycnogenol^®^ and placebo group at three time points (baseline, after 2 months, and after 3 months). Data are presented as mean with standard deviation and standardized to 13 participants, as 13 was the maximum number of participants in the Pycnogenol^®^ group with the metabolite M1 in saliva. In the placebo group, M1 was detectable in only 2 participants. Significant differences are indicated by asterisks (*), with a significance level of α < 0.05.

**Figure 3 nutrients-17-01546-f003:**
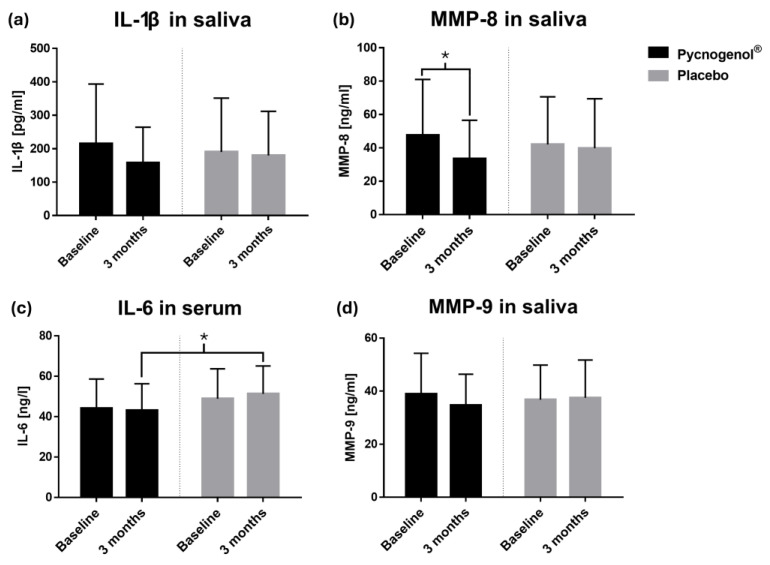
Concentrations at two time points (baseline and after 3 months) of (**a**) IL-1β in saliva, n(Pycnogenol^®^) = 35, n(Placebo) = 38; (**b**) MMP-8 in saliva, n(Pycnogenol^®^) = 37, n(Placebo) = 37; (**c**) IL-6 in serum, n(Pycnogenol^®^) = 38, n(Placebo) = 40; and (**d**) MMP-9 in saliva, n(Pycnogenol^®^) = 37, n(Placebo) = 38. Data are presented as mean with standard deviation. Significant differences are indicated by asterisks (*), with a significance level of α < 0.05.

**Figure 4 nutrients-17-01546-f004:**
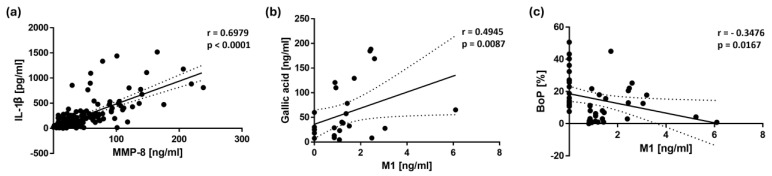
Scatter plots illustrating the correlations with 95% confidence intervals between (**a**) MMP-8 and IL-1β concentrations in saliva, (**b**) M1 and gallic acid concentrations in saliva, and (**c**) M1 concentration in saliva and bleeding on probing (BoP). The Spearman’s rank correlation analysis revealed a strong positive association between MMP-8 and IL-1β (r = 0.6979, *p* < 0.0001, *n* = 144), as well as between M1 and gallic acid (r = 0.4945, *p* = 0.0087, *n* = 27). A significant inverse correlation was observed between M1 and BoP (r = −0.3476, *p* = 0.0167, *n* = 47). Each point represents an individual data pair.

**Table 1 nutrients-17-01546-t001:** Demographic and lifestyle characteristics.

Variables	Pycnogenol^®^ Group (*n* = 46)	Placebo Group (*n* = 45)	*p*-Value ^1^
Age (years), mean ± SD	67.83 ± 9.18	66.96 ± 8.20	0.6348 (T)
Gender, *n* (%)			
Male	27 (58.7)	12 (26.7)	0.0029 * (F)
Female	19 (41.3)	33 (73.3)	
Vegetarian diet, *n* (%)			
Yes	6 (13.0)	2 (4.4)	0.2668 (F)
No	40 (87.0)	43 (95.6)	
Smoking status, *n* (%)			
Yes	5 (10.9)	2 (4.4)	0.4345 (F)
No	41 (89.1)	43 (95.6)	
Drinking status ^2^, *n* (%)			
Yes	22 (47.8)	15 (33.3)	0.2018 (F)
No	24 (52.2)	30 (66.7)	

^1^ Compared between groups; two-tailed *t*-test (T), Fisher’s exact test (F). ^2^ At least once a week. * Significant *p*-value for α < 0.05.

**Table 2 nutrients-17-01546-t002:** Concentrations of specific polyphenols in saliva and biomarker levels in saliva and serum, measured in the Pycnogenol^®^ and placebo group during the study.

Analyte	Measurement Time	Pycnogenol^®^		Placebo		*p*-Value ^b^	*p*-Value ^c^	*p*-Value ^d^
		Mean ± SD	*n* ^1^	Mean ± SD	*n* ^1^			
Taxifolin [ng/mL]	Baseline (Visit 1)	3.46 ± 2.14	5	0.00	0	0.0357 *		0.0257 *
	2 months (Visit 2)	5.57 ± 6.12	8	4.47 ± 2.07	4	0.5697		0.2258
	*p*-Value ^a^	0.3750		0.1250				
	3 months (Visit 3)	3.29 ± 2.46	5	2.89 ± 1.39	5	0.8413	0.1229	1.000
	*p*-Value ^a^	0.7695		0.0625				
Ferulic acid [ng/mL]	Baseline (Visit 1)	10.28 ± 6.76	11	6.38 ± 2.85	9	0.3410		0.6135
	2 months (Visit 2)	18.47 ± 36.97	17	7.89 ± 7.53	7	0.9014		0.0155 *
	*p*-Value ^a^	0.2774		0.4131				
	3 months (Visit 3)	12.87 ± 16.23	17	7.56 ± 4.24	5	0.8912	0.5775	0.0027 *
	*p*-Value ^a^	0.2935		0.4697				
Caffeic acid [ng/mL]	Baseline (Visit 1)	17.70 ± 21.49	19	22.90 ± 26.34	9	0.9912		0.0201 *
	2 months (Visit 2)	28.12 ± 28.84	28	11.30 ± 15.34	22	0.0077 *		0.1714
	*p*-Value ^a^	0.0002 *		0.2726				
	3 months (Visit 3)	27.71 ± 30.99	21	16.38 ± 17.83	17	0.9364	0.0388 *	0.3766
	*p*-Value ^a^	0.1908		0.3209				
Gallic acid [ng/mL]	Baseline (Visit 1)	45.59 ± 50.04	9	89.82 ± 66.88	3	0.4818		0.0667
	2 months (Visit 2)	82.27 ± 84.34	15	68.69 ± 88.27	11	0.3051		0.3474
	*p*-Value ^a^	0.0046 *		0.0020 *				
	3 months (Visit 3)	45.32 ± 52.92	14	63.92 ± 66.28	8	0.2973	0.0102 *	0.1394
	*p*-Value ^a^	0.4037		0.1641				
p-Coumaric acid [ng/mL]	Baseline (Visit 1)	3.99 ± 2.92	22	5.09 ± 6.59	22	0.4678		1.000
	2 months (Visit 2)	6.71 ± 8.75	25	4.26 ± 3.27	28	0.9472		0.6445
	*p*-Value ^a^	0.3369		0.3058				
	3 months (Visit 3)	7.93 ± 15.43	29	4.41 ± 4.42	25	0.8063	0.2630	0.3774
	*p*-Value ^a^	0.1827		0.9911				
Protocatechuic acid [ng/mL]	Baseline (Visit 1)	24.17 ± 16.74	17	22.84 ± 26.36	22	0.4617		0.3766
	2 months (Visit 2)	48.01 ± 87.84	20	31.19 ± 36.18	24	0.5360		0.5067
	*p*-Value ^a^	0.3596		0.2746				
	3 months (Visit 3)	35.47 ± 36.43	18	29.43 ± 29.43	22	0.4228	0.0954	0.5077
	*p*-Value ^a^	0.2292		0.9019				
M1 [ng/mL]	Baseline (Visit 1)	2.60	1	0.00	0	-		0.4938
	2 months (Visit 2)	1.67 ± 1.45	13	1.09	1	0.0929		0.0003 *
	*p*-Value ^a^	0.0005 *		-				
	3 months (Visit 3)	1.96 ± 1.28	13	1.37	1	0.0643	0.0252 *	0.0003 *
	*p*-Value ^a^	0.0024 *		-				
IL-1β [pg/mL]	Baseline (Visit 1)	214.69 ± 178.80	35	190.11 ± 161.39	38	0.4897		-
	3 months (Visit 3)	157.22 ± 107.14	35	179.88 ± 132.07	38	0.6072	0.1145	-
	*p*-Value ^a^	0.1347		0.6379				
MMP-8 [ng/mL]	Baseline (Visit 1)	47.65 ± 33.39	37	42.06 ± 28.58	37	0.5790		-
	3 months (Visit 3)	33.43 ± 23.15	37	39.78 ± 29.69	37	0.5209	0.0414 *	-
	*p*-Value ^a^	0.0261 *		0.6916				
MMP-9 [ng/mL]	Baseline (Visit 1)	38.91 ± 15.37	37	36.75 ± 13.09	38	0.4435		-
	3 months (Visit 3)	34.67 ± 11.69	37	37.48 ± 14.23	38	0.8187	0.1215	-
	*p*-Value ^a^	0.1633		0.8075				
IL-6 ^2^ [ng/mL]	Baseline (Visit 1)	44.20 ± 14.43	38	48.88 ± 14.83	40	0.1793		-
	3 months (Visit 3)	43.16 ± 13.17	38	51.27 ± 13.84	40	0.0409 *	0.2561	-
	*p*-Value ^a^	0.5326		0.3193				

^1^ Number of participants in whom the respective analyte could be quantified. Values below the LLOQ [18] were processed as 0 ng/mL to enable statistical testing. ^2^ Determined in serum, while all other analytes were determined in saliva. ^a^ Concentrations compared within group; *p*-value for Wilcoxon matched-pairs signed-rank test. ^b^ Concentrations compared between groups; *p*-value for Mann–Whitney test. ^c^ Concentrations compared between groups; *p*-value for repeated measures ANOVA. ^d^ Number of participants compared between groups; *p*-value for Fisher’s exact test. * Significant *p*-value for α < 0.05.

**Table 3 nutrients-17-01546-t003:** Spearman’s rank correlation coefficients (r) and *p*-values for the associations between the clinical parameter bleeding on probing (BoP), inflammation-related biomarkers and salivary polyphenol levels.

Variable 1	Variable 2	*n* ^1^	Spearman’s r	*p*-Value *
BoP	MMP-8	148	0.1906	0.0203
BoP	IL-1β	146	0.2380	0.0038
MMP-8	IL-1β	144	0.6979	<0.0001
M1	Gallic acid	27	0.4945	0.0087
M1	BoP	47	−0.3476	0.0167

^1^ Number of analyzed pairs for each correlation. * Significant *p*-value for α < 0.05.

## Data Availability

The individual pseudonymous data presented in this study are available upon request from the corresponding author due to privacy/ethical restrictions.

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
