# Peer review of "Impact of a Dietary Supplementation with French Maritime Pine Bark Extract Pycnogenol® on Salivary and Serum Inflammatory Biomarkers During Non-Surgical Periodontal Therapy—A Randomized Placebo-Controlled Double-Blind Trial"

_nutrients, 2025, doi:10.3390/nu17091546_

Round 1
Reviewer 1 Report
Comments and Suggestions for Authors
-
Gender Imbalance Between Groups
-
The Pycnogenol® group included a significantly higher proportion of male participants compared to the placebo group (58.7% vs. 26.7%, p = 0.0029). This unequal gender distribution may confound inflammatory marker levels and should be addressed in the statistical analysis or included as a limitation.
-
Recommendation: Consider adjusting the results for gender or performing subgroup analyses. At minimum, this imbalance must be explicitly acknowledged in the discussion.
-
-
Missing Clinical Outcomes in This Manuscript
-
The title and abstract refer to non-surgical periodontal therapy, yet clinical outcomes (e.g., probing depth, clinical loss) are not reported here and are said to be included in another manuscript.
-
Recommendation: Even if the clinical findings are detailed elsewhere, this manuscript should briefly summarize them or clearly clarify their absence to prevent confusion or perceived incompleteness.
-
-
Unclear Biological Relevance of M1 as a Marker
-
While M1 (a gut metabolite of polyphenols) was significantly elevated in the Pycnogenol® group and correlated with BoP, its biological function remains speculative in the paper.
-
Recommendation: The authors should be more cautious in interpreting M1 as a potential “bioactive effector” and clearly distinguish between compliance monitoring and mechanistic roles.
-
-
Statistical Reporting Gaps
-
Although multiple tests were applied, the choice of statistical methods is not always clearly justified, and multiple testing corrections (e.g., Bonferroni) are not mentioned.
-
Recommendation: Clarify whether correction for multiple comparisons was applied, and if not, explain why.
-
-
Interpretation of “Significant” vs. “Clinically Relevant”
-
The authors report statistically significant reductions in MMP-8 and IL-6, but the clinical significance of these relatively modest changes (e.g., ~5–10 ng/mL) is not discussed.
-
Recommendation: Address whether these biomarker shifts translate into meaningful clinical improvements or warrant follow-up studies.
-
Reviewer 2 Report
Comments and Suggestions for Authors
please see enclosed pdf

Round 2
Reviewer 2 Report
Comments and Suggestions for Authors
The manuscript has been improved
Author Response
Answer Comment 1:
Thank you for your comment. In Table 2, "n" does not refer to the total sample size of the respective group, but rather to the number of participants for whom the respective analyte could be quantified. This definition of "n" is also provided in the table’s footnote. As outlined in the CONSORT statement, the statistical analysis was conducted based on the actual number of participants available at each time point. For example, in the placebo group, data from 45 participants were analyzed at visit 1, 42 at visit 2, and 41 at visit 3.
Answer Comment 2:
Thank you for your valuable suggestion. To improve clarity and focus on the most relevant correlations, we propose to simplify Table 3 accordingly. We will retain the correlations between the dependent clinical parameter BoP and the respective biomarkers and polyphenols, as well as include important biomarker-biomarker associations that are critical for the interpretation of the inflammatory response (e.g., MMP-8 and IL-1β). Thus, the revised Table 3 will present the correlations for BoP vs. MMP-8, BoP vs. IL-1β, M1 vs. BoP, M1 vs. Gallic acid, and MMP-8 vs. IL-1β.